# ConceptMoE: Adaptive Token-to-Concept Compression for Implicit Compute Allocation

**Zihao Huang** [* 1] **Jundong Zhou** [* 1] **Xingwei Qu** [* 1] **Qiyang Min** [* 1] **Ge Zhang** [1]

## Abstract

Large language models allocate uniform computation across all tokens, ignoring that some sequences are trivially predictable while others require deep reasoning. We introduce ConceptMoE, which dynamically merges semantically similar tokens into concepts through learnable chunking at target compression ratio $R$. The MoE architecture enables controlled evaluation: reallocating saved computation to match baseline FLOPs and parameters isolates genuine architectural benefits. ConceptMoE consistently outperforms standard MoE, achieving +0.9 points on language pretraining, +2.3 on long context, and +0.6 on multimodal tasks. Continual training conversion with layer looping gains +5.5 points. Beyond performance, at $R = 2$, ConceptMoE reduces attention computation by $R^2 \times$ and KV cache by $R \times$, delivering prefill speedups up to 175% and decoding speedups up to 117%. The minimal architectural changes enable straightforward integration, demonstrating that adaptive concept-level processing fundamentally improves LLM effectiveness and efficiency.

## 1. Introduction

Large language models process every token with equal computational effort, ignoring that understanding naturally operates at varying granularities. Humans don't process "New", "York", and "City" separately, we recognize the unified concept "New York City" instantly. Yet LLMs waste computation dissecting trivially predictable tokens while allocating insufficient resources to semantically dense positions requiring deep reasoning. This mismatch between uniform token-level computation and hierarchical semantic structure fundamentally limits both efficiency and effectiveness.

The field has long sought to compress token sequences while preserving semantics. Vocabulary expansion offers one path: larger vocabularies compress text into fewer, information-denser tokens (Takase et al., 2025). However, gains are limited, a $100 \times$ vocabulary increase yields only $1.3 \times$ compression, and massive vocabularies create computational bottlenecks during training and inference.

Dynamic token merging within models presents an alternative paradigm. Rather than processing at fixed token granularity, we can adaptively identify semantic units and process at concept level. Several approaches (Dai et al., 2025; Shao et al., 2025) merge consecutive tokens into concepts, but employ rigid fixed-length or rule-based strategies. Byte-level models (Yu et al., 2023; Pagnoni et al., 2024; Hwang et al., 2025) explore adaptive chunking, yet lack precise parameter control and introduce confounding factors through representation changes. Most critically, evaluations typically compensate reduced tokens by scaling model size, conflating architectural innovation with brute-force parameter increases.

We introduce ConceptMoE, which fundamentally shifts processing from token level to concept level through learnable adaptive chunking. The core mechanism is intuitive: consecutive tokens with high semantic similarity merge into unified concept representations that the model processes jointly, while semantically distinct tokens maintain fine-grained granularity. A learnable chunk module identifies optimal boundaries by measuring inter-token similarity, compressing sequences before they enter the compute-intensive concept model. This realizes implicit compute allocation, the model automatically invests computation where semantic complexity demands it while efficiently handling predictable patterns.

Crucially, the MoE architecture enables rigorous controlled evaluation. By reallocating computation saved from compression to match baseline FLOPs and total parameters, we isolate genuine architectural benefits from concept-level processing. Our contributions demonstrate this paradigm shift: (1) Fair comparison under identical parameters and FLOPs across three compute strategies, showing consistent improvements. (2) Versatile deployment: language pretraining (+0.9 points), vision-language training with dual-

---

[*]Equal contribution [1]Bytedance Seed. Correspondence to: Ge Zhang <zhangge.eli@bytedance.com>.

*Proceedings of the $43^{rd}$ International Conference on Machine Learning*, Seoul, South Korea. PMLR 306, 2026. Copyright 2026 by the author(s).

modality compression (+0.6 points, +2.3 on long context), and lossless continual training conversion (+5.5 points, +6.4 from scratch). (3) Inherent efficiency: at compression ratio $R$, attention computation reduces by $R^2\times$ and KV cache by $R\times$, achieving prefill speedups up to 175% and decoding speedups up to 117%. (4) Practical integration: minimal architectural changes enable straightforward deployment in existing systems.

ConceptMoE represents the next paradigm for language models: adaptive concept-level processing that aligns computational effort with semantic structure, fundamentally improving both effectiveness and efficiency.

## 2. Related Work

**Vocabulary size** determines the information capacity of each token in LLMs, typically ranging from 32K to 256K. Larger vocabularies enable higher compression efficiency and fewer tokens per sequence. Tao et al. (2024) demonstrate that vocabulary size should scale with model parameters, while Takase et al. (2025) show that increasing vocabulary from 5K to 500K achieves a compression ratio of 1.3, yielding substantial downstream improvements when training on fixed data or token budgets. However, a $100\times$ vocabulary expansion produces only $1.3\times$ compression, revealing that further compression gains require exponential vocabulary growth. Moreover, excessively large vocabularies become inference bottlenecks (Cai et al., 2024; Liu et al., 2025). This motivates an alternative approach: dynamic compression within the model itself, which forms the foundation of ConceptMoE.

**Token-level chunking** compresses tokens internally within the model. Existing approaches include fixed-length merging (Dai et al., 2025; Shao et al., 2025) and heuristic or rule-based compression (Ankireddy et al.; Geng et al., 2025). These methods merge multiple tokens into single concepts to achieve higher compression ratios without expanding vocabulary size, but lack adaptive compression strategies. Token information density varies significantly: information-sparse tokens should merge aggressively, while information-rich tokens should maintain finer granularity. Recent concurrent work DLCM(Qu et al., 2025) introduces dynamic compression to merge tokens into concepts, but when comparing against FLOPs-matched baselines, the model parameters actually double, undermining the fairness of ablation studies. ConceptMoE addresses these through a learnable chunk module that adaptively identifies optimal merge boundaries based on semantic similarity. Crucially, we leverage MoE properties to ensure all comparisons maintain identical total parameters and average per-token FLOPs, enabling rigorous ablation studies.

**Byte-level models** require more sophisticated merging

strategies due to the granularity of byte tokens, which necessitates aggressive compression to maintain manageable computational costs. AU-Net (Videau et al., 2025) introduces multi-level compression but relies on rule-based strategies. BLT (Pagnoni et al., 2024) leverages a pretrained auxiliary model to compute token entropy, merging low-entropy tokens more aggressively. However, this non-end-to-end approach may encounter difficulties during downstream fine-tuning. H-Net (Hwang et al., 2025) proposes an end-to-end dynamic chunking module, achieving $9\times$ compression at the byte level. While this represents approximately $2\times$ compression at the token level, H-Net's experiments control only FLOPs while allowing total parameters to vary, introducing confounding factors. Additionally, the byte-level input representation itself constitutes an experimental variable. ConceptMoE builds on insights from H-Net but operates at the token level with higher compression efficiency and provides fair comparison by controlling both FLOPs and total parameters. We validate effectiveness and efficiency at significantly larger model scales across diverse training scenarios, including language-only pretraining, vision-language training, and continual training from pretrained checkpoints.

## 3. Approach

### 3.1. Overview

Just like byte level models, our ConceptMoE is also composed of 5 modules, namely the encoder $\mathcal{E}$, chunk module Chunk, concept model $\mathcal{C}$, dechunk module DeChunk, and decoder $\mathcal{D}$, as shown in Figure 1. $\mathcal{E}$, $\mathcal{C}$ and $\mathcal{D}$ are all composed of multiple stacked MoE modules. Given a sequence of input hidden state $\boldsymbol{H} = \{\boldsymbol{h}_1, \boldsymbol{h}_2, \ldots, \boldsymbol{h}_n, \ldots, \boldsymbol{h}_N\}$, $\boldsymbol{h}_n \in \mathbb{R}^d$,

$$\hat{\boldsymbol{H}} = \mathcal{E}(\boldsymbol{H}), \quad \boldsymbol{C}, \boldsymbol{P} = \mathsf{Chunk}(\hat{\boldsymbol{H}}),$$
$$\hat{\boldsymbol{C}} = \mathcal{C}(\boldsymbol{C}), \quad \boldsymbol{Z} = \mathsf{DeChunk}(\hat{\boldsymbol{C}}, \boldsymbol{P}), \quad (1)$$
$$\hat{\boldsymbol{Z}} = \mathcal{D}(\hat{\boldsymbol{C}}, \boldsymbol{Z}),$$

where $\boldsymbol{C} = \{\boldsymbol{c}_1, \boldsymbol{c}_2, \ldots, \boldsymbol{c}_m, \ldots, \boldsymbol{c}_M\}$, $\boldsymbol{c}_m \in \mathbb{R}^{d1}$ is the concept embeddings, $M \leq N$, $\hat{\boldsymbol{H}}$, $\boldsymbol{Z}$, $\hat{\boldsymbol{Z}}$ and $\boldsymbol{H}$ have the same dimension. $\boldsymbol{P} = \{p_1, p_2, \ldots, p_n, \ldots, p_N\}$ is the probability that the token is a chunk boundary. Note that in the $\mathcal{D}$ input, each token is guaranteed to have an associated concept for joint decoding. This significantly enhances model capability by ensuring that information in the concept is fully utilized across multiple subsequent tokens.

In general, the computational proportion of $\mathcal{E}$ and $\mathcal{D}$ is relatively small, and the main FLOPs come from the $\mathcal{C}$. The Chunk will select to merge multiple consecutive $\hat{\boldsymbol{h}}$

---

[1]For the sake of simplicity, the dimensions of $\boldsymbol{c}$ and $\boldsymbol{h}$ are kept consistent here, both being $d$; in fact, the dimension of $\boldsymbol{c}$ can be larger than that of $\boldsymbol{h}$.

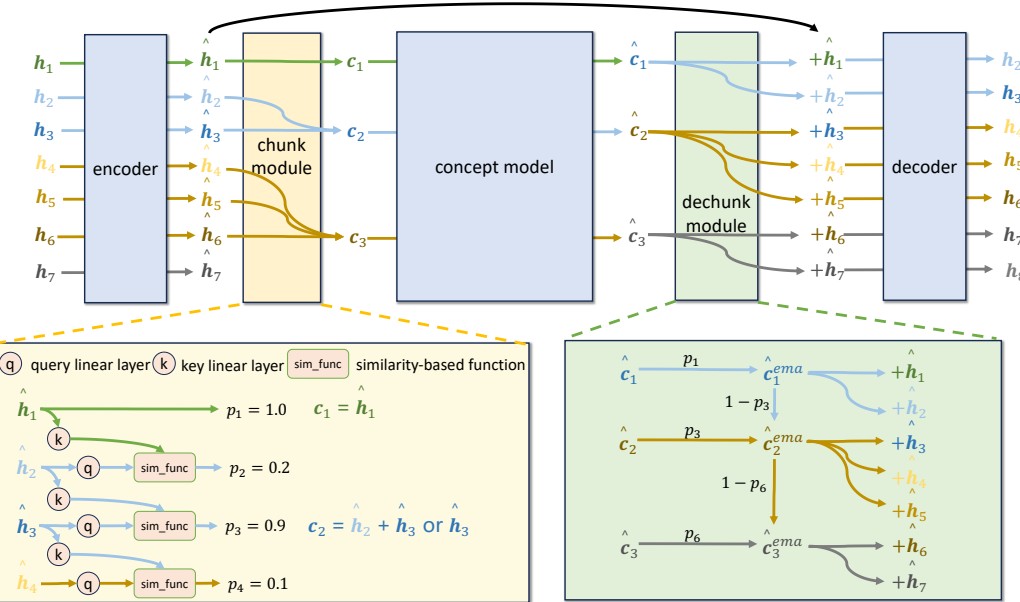

*Figure 1.* Overview of ConceptMoE, with details of chunk and dechunk modules.

into single $c$ to enter $\mathcal{C}$. We regard the merged multiple tokens as a concept, hence the intermediate model is called the concept model. We provide PyTorch-like code in the Appendix D for better understanding.

### 3.2. Chunk module

The chunk module aims to identify optimal chunk boundaries for a given input token sequence under a specified compression ratio. Similar to H-Net(Hwang et al., 2025), it selects which consecutive tokens are easily predictable and merges them into chunks, while keeping hard-to-predict tokens unmerged or merged into smaller chunks. This approach performs implicit **token-level compute allocation**, in contrast to explicit compute allocation methods like MoE (e.g., Zero Expert(Jin et al., 2024)) that activate different amounts of computation for each token.

Specifically, for the input $\boldsymbol{h}_n$, we calculate the cosine similarity between the embeddings of two adjacent tokens after linear transformation to determine if this token serves as a chunk boundary:

$$\boldsymbol{q}_n = W_q \boldsymbol{h}_n, \quad \boldsymbol{k}_n = W_k \boldsymbol{h}_n,$$

$$p_n = \frac{1}{2}\Big(1 - \underbrace{\frac{\boldsymbol{q}_n^\top \boldsymbol{k}_{n-1}}{\|\boldsymbol{q}_n\| \cdot \|\boldsymbol{k}_{n-1}\|}}_{\text{cosine}}\Big), \quad b_n = \mathbb{1}_{p_n \geq 0.5}, \quad (2)$$

where $W_q, W_k \in \mathbb{R}^{d \times d}$ are learnable parameters, and we set $p_1 = 1.0$ to ensure that the first token in the decoder $\mathcal{D}$ always has a concept for joint decoding. If the probability $p_n$ exceeds 0.5, the token is identified as a chunk

boundary. This design is natural and intuitive: when several consecutive tokens exhibit high similarity but the current token shows low similarity to its predecessor, it indicates a significant semantic shift. Such tokens typically carry substantially different information and require more careful processing.

**Auxiliary loss** Given a target compression ratio, we introduce an auxiliary loss to constrain the token compression ratio on the training set, where the compression ratio is defined as $R = N/M \geq 1$. Inspired by the load balancing loss in MoE, we treat boundary and non-boundary selections as two experts and constrain their activation frequencies to achieve the target compression ratio. Let the average probabilities and average selection ratios for boundary and non-boundary be:

$$G_1 = \frac{1}{N}\sum_{n=1}^{N} p_n, \quad G_2 = 1 - G_1 = \frac{1}{N}\sum_{n=1}^{N}(1 - p_n),$$

$$F_1 = \frac{1}{N}\sum_{n=1}^{N} b_n, \quad F_2 = 1 - F_1 = \frac{1}{N}\sum_{n=1}^{N}(1 - b_n). \quad (3)$$

then the auxiliary loss is

$$\begin{aligned}
\mathcal{L}_{\text{aux}} &= RF_1G_1 + \frac{R}{R-1}F_2G_2 \\
&= \frac{R}{R-1}\Big[(R-1)F_1G_1 + (1-F_1)(1-G_1)\Big].
\end{aligned} \quad (4)$$

Note that when computing $G_1$ and $F_1$, we aggregate statistics across all samples in the current devices rather than averaging per-sample statistics. This enables **sample-level**

**compute allocation**: when a batch contains both difficult and easy samples, the model can reduce the compression ratio for difficult samples while increasing it for easy samples. In practical use, we use $\lambda$ to control the weight of this auxiliary loss.

**Random flip boundary** When we constrain the compression ratio on the training set, empirical observations show that distribution shifts in the evaluation set can lead to excessively high compression ratio, which speeds up inference but significantly degrades model performance. We address this by adding random perturbations to boundaries during training to simulate such scenarios, thereby mitigating the over-compression issue caused by distribution shifts at inference time. Specifically, after computing $p_n$, we sharpen the probability:

$$p_n^{sharp} = \begin{cases} p_n^{\frac{1}{\tau}} & p_n \geq 0.5 \\ 1 - (1-p_n)^{\frac{1}{\tau}} & p_n < 0.5 \end{cases} \quad (5)$$

Where $\tau$ is a hyperparameter to control sharpness. The final boundary is obtained by sampling from a Bernoulli distribution parameterized by $p_n^{sharp}$:

$$b_n^{\text{train}} \sim \text{Bernoulli}(p_n^{\text{sharp}}), \ b_n^{\text{train}} \in \{0, 1\},$$

$$\Pr(b_n^{\text{train}} = k) = \begin{cases} p_n^{\text{sharp}}, & k = 1, \\ 1 - p_n^{\text{sharp}}, & k = 0. \end{cases} \quad (6)$$

The sharpened probability distribution is more likely to flip boundaries with low confidence (close to 0.5) during sampling, while probabilities with high confidence (close to 0 or 1) are less likely to be flipped. This ensures stable training convergence.

**Merging strategy** We can either sum all tokens within a chunk to obtain the concept, which maximally preserves the information of each token, or use only the last token as the concept, since the self-attention mechanism in the encoder $\mathcal{E}$ enables the last token to already aggregate the information of the entire chunk.

### 3.3. Dechunk module

The dechunk module remaps concepts back to tokens and ensures that no information leakage occurs. Before dechunk, we first apply exponential moving average (EMA) on concepts. Let $\mathcal{I} = \{n \mid b_n = 1, 1 \leq n \leq N\}$ with $|\mathcal{I}| = M$. Define the index mapping

$$\phi : \{1, 2, \ldots, m, \ldots, M\} \to \mathcal{I}, \quad \phi(m) = n_m, \quad (7)$$

then for each $\hat{c}_m$, there is

$$\hat{c}_m^{ema} = p_{\phi(m)}\hat{c}_m + (1 - p_{\phi(m)})\hat{c}_{m-1}. \quad (8)$$

This mechanism is illustrated in the green part of Figure 1. The EMA accelerates chunking convergence through the

following process: consider an example where "Simple and easy-to-" and "understand picture" are initially split into two concepts, with $p = 0.5$ for "understand picture". Through EMA, if the model discovers that the concept of "Simple and easy-to-" effectively aids in predicting tokens within "understand picture", it will reduce the boundary probability $p$ for "understand picture". Once $p$ falls below 0.5, the boundary is eliminated, merging "Simple and easy-to-" and "understand picture" into a unified concept.

After EMA, based on the given index mapping, dechunk the concept to the token-level embedding. Specifically, we define the mapping from concept index to token index:

$$\psi : \{1, \ldots, N\} \to \{1, \ldots, M\},$$
$$\psi(n) = m \quad \text{if } \phi(m) \leq n < \phi(m+1). \quad (9)$$

Finally we can obtain the $\mathcal{D}$ input $z_n$ as

$$\boldsymbol{z_n} = \hat{\boldsymbol{h}}_n + \hat{\boldsymbol{c}}_{\psi(n)}^{ema}. \quad (10)$$

### 3.4. Joint decoding

During token decoding, multiple tokens share the same concept in the dechunking process. To fully exploit the information in the concept, which accounts for most of the model's computation and contains rich information, we perform joint decoding of $\boldsymbol{z_n}$ and $\hat{\boldsymbol{c}}_{\psi(n)}^{ema}$. Specifically, in each self-attention layer of the decoder $\mathcal{D}$, we have:

$$\tilde{\boldsymbol{q}}_n = \boldsymbol{z}_n W_q + \hat{\boldsymbol{c}}_{\psi(n)}^{\text{ema}} W_q^c,$$
$$\tilde{\boldsymbol{k}}_n = \boldsymbol{z}_n W_k + \hat{\boldsymbol{c}}_{\psi(n)}^{\text{ema}} W_k^c,$$
$$\tilde{\boldsymbol{v}}_n = \boldsymbol{z}_n W_v + \hat{\boldsymbol{c}}_{\psi(n)}^{\text{ema}} W_v^c,$$
$$\text{Attention}(\boldsymbol{z}_n, \hat{\boldsymbol{c}}_{\psi(n)}^{\text{ema}}) = \text{softmax}\left(\frac{\tilde{\boldsymbol{q}}_n \tilde{\boldsymbol{k}}_n^T}{\sqrt{d_{\text{head}}}} + M\right) \tilde{\boldsymbol{v}}_n.$$
$$(11)$$

where $M$ is the causal attention mask with $M_{ij} = -\infty \cdot \mathbb{1}_{i<j}$. The highlighted terms show how we augment standard attention by incorporating concept information into the query, key, and value. Given the shallow depth of decoder $\mathcal{D}$, the additional parameters are negligible while yielding substantial performance gains. Moreover, this design maintains compatibility with existing architectures, enabling straightforward application to continue training (CT) from pretrained models.

### 3.5. Compute reallocation strategies for fair comparison

Given a standard MoE model with encoder, concept model, and decoder of depths $L_{\mathcal{E}}$, $L_{\mathcal{C}}$, and $L_{\mathcal{D}}$, let $C_{attn}$ and $C_{moe}$ denote FLOPs per token in self-attention and MoE layers (excluding attention maps). The concept model incurs $L_{\mathcal{C}}(C_{attn} + C_{moe})$ FLOPs per token. At compression ratio $R$, the encoder outputs $R$ tokens before the con-

cept model processes one concept, reducing computation to $\frac{L_{\mathcal{C}}(C_{attn}+C_{moe})}{R}$.

We reallocate this saved computation by increasing $L_{\mathcal{C}}$, $C_{attn}$, or $C_{moe}$ to maintain total FLOPs. Unlike dense models, MoE allows adjusting activated parameters while fixing total parameters, enabling fair comparison: **under identical total parameters and per-token FLOPs, we isolate ConceptMoE's true gains by varying only activated parameters.** We explore three strategies:

1. **Increasing $C_{moe}$**: Activate more experts per token. Simple and applicable to both pretraining and continual training.

2. **Increasing $L_{\mathcal{C}}$ and $C_{moe}$**: Activate more experts plus loop intermediate layers. CT-friendly with no additional parameters.

3. **Increasing $C_{attn}$ and $C_{moe}$**: Scale hidden size while reducing total experts, requiring two projectors for $\hat{h} \rightarrow c$ and $\hat{c} \rightarrow z$. Better for pretraining than CT.

Beyond matched FLOPs, ConceptMoE inherently reduces attention map computation and KV cache. For standard MoE, the concept model computes attention maps costing $L_{\mathcal{C}}dN^2$ FLOPs and stores KV cache of $2L_{\mathcal{C}}dN$, where $d$ is hidden size and $N$ is sequence length. Table 1 shows reductions for each strategy (where $L_{loop}$ is the number of looped layers). All strategies substantially reduce both metrics.

*Table 1.* Attention-map FLOPs & KV-cache comparison.

| Method | Attn FLOPs | Reduction | KV-cache | Reduction |
|---|---|---|---|---|
| Baseline (MoE) | $L_{\mathcal{C}}dN^2$ | $1\times$ | $2L_{\mathcal{C}}dN$ | $1\times$ |
| $C_{\text{moe}}$ | $\frac{L_{\mathcal{C}}dN^2}{R^2}$ | $R^2\times$ | $\frac{2L_{\mathcal{C}}dN}{R}$ | $R\times$ |
| $L_{\mathcal{C}}$ & $C_{\text{moe}}$ | $\frac{(L_{\mathcal{C}}+L_{\text{loop}})dN^2}{R^2}$ | $\frac{R^2 L_{\mathcal{C}}}{L_{\mathcal{C}}+L_{\text{loop}}}\times$ | $\frac{2(L_{\mathcal{C}}+L_{\text{loop}})dN}{R}$ | $\frac{RL_{\mathcal{C}}}{L_{\mathcal{C}}+L_{\text{loop}}}\times$ |
| $C_{\text{attn}}$ & $C_{\text{moe}}$ | $\frac{L_{\mathcal{C}}dN^2}{R^{1.5}}$ | $R^{1.5}\times$ | $\frac{2L_{\mathcal{C}}dN}{\sqrt{R}}$ | $\sqrt{R}\times$ |

## 4. Experiments

We conduct comprehensive experiments to validate ConceptMoE's effectiveness and efficiency across diverse settings. Our evaluation encompasses four main aspects: (1) Small-scale language model pretraining (Section 4.1) at 12B and 24B parameters to establish core benefits under controlled conditions. (2) Vision-language model training (Section 4.2) at 60B parameters, demonstrating dual-modality compression and achieving strong gains on long context tasks. (3) Continual training conversion (Section 4.3) from pretrained MoE at 90B parameters, validating practical deployment through lossless integration. (4) Inference speedup analysis (Section 4.4) on 300B parameters models, measuring actual latency improvements across diverse compression ratios and

layer configurations. Additionally, we perform extensive ablation studies examining auxiliary loss weight, chunking strategies, router design, joint decoding, boundary noise, and target compression ratios. All experiments maintain identical total parameters and per-token FLOPs (excluding attention maps) between ConceptMoE and MoE, ensuring fair architectural comparison.

**Common model configuration** The MoE baseline activates 8 experts. ConceptMoE uses an auxiliary loss weight of $\lambda = 0.03$. For token merging strategy, models with CT integration use only the last token in each chunk as the concept to minimize structural modifications when converting MoE to ConceptMoE and keep the initial loss as low as possible. Other models sum tokens within each chunk to form concepts. The hyperparameter for boundary random flipping is set to $\tau = 6$, under which approximately 4% of tokens are flipped. This configuration keeps the evaluation compression ratio close to the training compression ratio without affecting training performance.

**Evaluation Benchmarks** We conduct extensive evaluations on both open-source and proprietary benchmarks. The evaluation suite covers text reasoning, mathematics, code generation, knowledge retrieval, needle-in-haystack, long context summarization and understanding, as well as multimodal tasks including visual localization, visual reasoning, hallucination detection, visual question answering, and chart extraction. Detailed benchmark specifications are provided in the Appendix C.

### 4.1. Small-scale language model pretraining

**Model configurations** We evaluate ConceptMoE on two MoE configurations: a 0.5B FLOPs model with 12B total parameters (MoE-A0.5B-12B), and a 1B FLOPs model with 24B total parameters(MoE-A1B-24B). Both baselines activate 8 experts per token. For ConceptMoE, we set $L_{\mathcal{E}} = L_{\mathcal{D}} = 4$ and scale the hidden size of $\mathcal{C}$ to $4/3$ that of $\mathcal{E}$ and $\mathcal{D}$. This configuration increases the per-token compute of $\mathcal{C}$ by a factor of $16/9$ relative to the baseline. We therefore set the compression ratio to $R = 16/9$ to match the total compute budget [2]. This setup corresponds to the third reallocation strategy in Section 3.5, which jointly increases $C_{attn}$ and $C_{moe}$.

**Training Protocol** We define tokens per parameter (TPP) as the ratio of training tokens to total model parameters. All models are trained at TPP=400: MoE-A0.5B-12B on 243B tokens and MoE-A1B-24B on 559B tokens. We use the AdamW optimizer with cosine learning rate decay.

**Results** Table 2 presents the main results comparing Con-

---

[2]Note that all FLOPs comparisons exclude attention map computation, meaning ConceptMoE actually consumes fewer FLOPs than the baseline when compute-matched.

ceptMoE against standard MoE baselines. Both model pairs maintain identical total parameters and per-token FLOPs, differing only in compute allocation strategy under concept or token level process. ConceptMoE consistently outperforms the baseline across most metrics.

These results validate that concept-level processing provides genuine benefits beyond simple token-level processing, suggesting that adaptive chunking effectively allocates more compute to semantically complex token sequences while efficiently processing predictable patterns.

*Table 2.* Model performance comparison. CMoE: ConceptMoE, **S**: Activate 0.5B with total 12B parameters, **L**: Activate 1B with total 24B parameters. Loss↓, others↑. Cmp: Comprehensive Evaluation, Reason: Reasoning, Know: Knowledge. Details see Table 8.

| Model | Train | | Eval | Openbench Easy | | | | | |
|---|---|---|---|---|---|---|---|---|---|
| | Token | Loss | Loss | Cmp | Reason | Math | Code | Know | All |
| MoE-S | 243B | 1.852 | 1.992 | 46.2 | 37.6 | 27.8 | 30.7 | **26.2** | 35.6 |
| CMoE-S | 243B | **1.849** | **1.990** | **47.3** | **39.1** | **28.8** | 30.7 | 26.1 | **36.4** |
| MoE-L | 559B | 1.717 | 1.851 | **57.6** | 54.4 | 47.2 | 42.3 | **41.6** | 50.0 |
| CMoE-L | 559B | **1.711** | **1.844** | 57.4 | **56.8** | **50.0** | **42.4** | 41.5 | **50.9** |

## 4.2. Train a vision-language model

**Model configurations** We investigate the potential of concept representations as inputs for vision-language model (VLM). We conduct experiments on MoE-A2.5B-60B and ConceptMoE-A2.5B-60B. The vision encoder uses a small ViT(Dehghani et al., 2023) for image feature extraction, and a linear projection to align the dimensions of the visual token with the text tokens. On the LLM side, we maintain $L_{\mathcal{E}} = L_{\mathcal{D}} = 4$ and apply the third reallocation strategy from Section 3.5. We scale the hidden size of $\mathcal{C}$ to 1.5 while slightly reducing the MoE inner dimension, and set the compression ratio to $R = 2$ to preserve total activated compute. Note that in the VLM setting, we apply compression to both visual and textual tokens.

**Training Protocol** We first pretrain the LLM on 500B tokens, then integrate a pretrained NaViT and a randomly initialized linear projection layer. The VLM continues training for 200B tokens on a mixture of high-quality image-text pairs and pure text data with 32K sequence length. Training uses the AdamW optimizer with cosine learning rate decay throughout.

**Results** Figure 2 shows training dynamics. During PT (a), ConceptMoE achieves 0.01 lower loss than MoE. In multimodal CT (b), this gap increases to 0.017 for text data and 0.012 for image-text data. This difference reflects adaptive compression (c): while maintaining overall $R = 2$, the model compresses text less (using more compute) and images more, suggesting higher visual redundancy.

Tables 3, 4, and 5 present downstream results with average compression ratios of 2.27, 2.0, and 2.0 respectively. ConceptMoE outperforms MoE by 0.9 points on text benchmarks, 2.3 on long context, and 0.6 on multimodal tasks. Long context shows particularly strong gains across most subtasks, as expected from reduced sequence length alleviating degradation on long documents. The Needle task improvement confirms that concept merging preserves information. Multimodal results show gains in reasoning and understanding, validating stronger concept-level reasoning. However, fine-grained visual tasks (location, chart Q&A, image Q&A) decline slightly, likely because treating image tokens sequentially disrupts spatial relationships critical for localization.

*Table 3.* Text benchmark results on OpenBench. Comp.: Comprehensive Evaluation, Spec.: Specialized Knowledge, Know.: Knowledge. Details see Table 9.

| Model | Comp. | Reason. | Math | Code | Spec. | Know. | All |
|---|---|---|---|---|---|---|---|
| MoE-A2.5B-60B | 41.7 | 35.0 | 16.3 | 34.6 | 28.8 | 35.0 | 33.5 |
| ConceptMoE-A2.5B-60B | **41.8** | **35.3** | 16.2 | **36.9** | 30.8 | **36.8** | **34.4** |

*Table 4.* Long context evaluation results. Needle: Needle in the haystack, LC: Long context, Learn.: Learning, Reason.: Reasoning, Sum.: Summary, Undrst.: Understanding.

| Model | Needle ↑ | LC Learn.↑ | LC Reason.↑ | LC Sum. /Q&A↑ | LC Undrst.↑ | All ↑ |
|---|---|---|---|---|---|---|
| MoE-A2.5B-60B | 78.9 | 34.7 | 10.3 | **63.7** | 40.9 | 49.4 |
| ConceptMoE-A2.5B-60B | **80.7** | **38.8** | **17.1** | 59.0 | **45.1** | **51.7** |

*Table 5.* Vision-language benchmark results. Loc.: Location, Reason.: Reasoning, Desc. Halluc.: Description Hallucinations, Q&A: Image Question and Answer, Comp.: Comprehensive Bench.

| Model | Loc. ↑ | Reason. ↑ | Desc. Halluc.↑ | Q&A ↑ | Chart Q&A↑ | Comp. ↑ | All ↑ |
|---|---|---|---|---|---|---|---|
| MoE-A2.5B-60B | **82.2** | 53.5 | **35.8** | **61.4** | **34.7** | 53.9 | 53.6 |
| ConceptMoE-A2.5B-60B | 81.9 | **57.9** | 35.3 | 58.0 | 33.6 | **58.7** | **54.2** |

## 4.3. Train from CT

In this subsection, we investigate converting MoE to ConceptMoE during continual training and evaluate the performance gap compared to training from scratch. We observe three key findings:

- **Lossless and beneficial CT conversion.** ConceptMoE-top15 matches baseline, while adding layer loops gains +5.5 points. From-scratch training reaches +6.4 points overall.

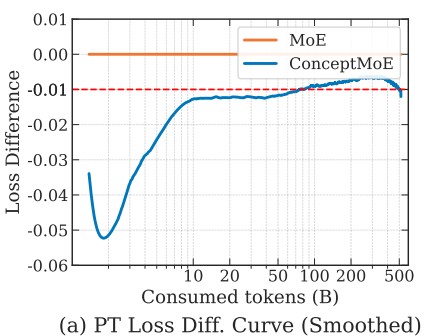
(a) PT Loss Diff. Curve (Smoothed)

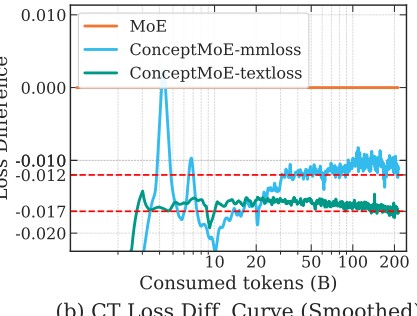
(b) CT Loss Diff. Curve (Smoothed)

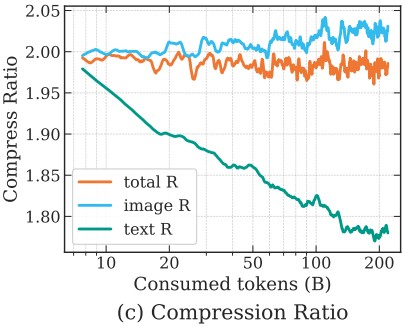
(c) Compression Ratio

*Figure 2.* Training dynamics of loss and compression ratio. (a) Loss difference between ConceptMoE and MoE during language model pretraining (PT). (b) Loss difference during multimodal continue training(CT), separated into image-text data (mmloss) and text-only data(textloss). (c) Compression ratio evolution for image tokens and text tokens during multimodal training.

- **Distribution shift effects.** PT evaluation shows higher compression ($R = 1.81$) due to data distribution mismatch, but CT data aligns with evaluation, stabilizing at target $R = 1.5$.

- **Direct inference speedup.** Stable $R = 1.5$ enables lossless conversion with significant gains: prefill up to 43.6% and decoding up to 53.3% faster (Section 4.4).

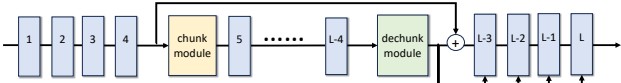

*Figure 3.* Converting MoE to ConceptMoE. Blue blocks are original MoE components. We add chunk and dechunk modules, plus QKV projectors (zero-initialized) in the last 4 self-attention layers for joint decoding.

Figure 3 illustrates the conversion process. Starting with MoE-A2.5B-90B pretrained on 700B tokens, we add a chunk module (random-initialized query/key layers), dechunk module, and QKV projectors in the final 4 layers. We perform 400B tokens of 32k CT, 40B of 128k CT, and 3B of SFT with math/reasoning problems. At conservative $R = 1.5$, we explore two strategies: ConceptMoE-top15 (15 activated experts, Strategy 1 in Section 3.5) and ConceptMoE-top11-loop8 (11 experts plus 8 looped layers, Strategy 2). Both match baseline FLOPs and parameters. We also train ConceptMoE-top11-loop8 from scratch for comparison.

Figure 4 shows evaluation dynamics. During CT (700B-1100B tokens), ConceptMoE-top15 matches MoE closely (0.3 point gap), while ConceptMoE-top11-loop8 clearly improves, confirming that reallocating compute to $L_C$ beats simply increasing $C_{moe}$. From-scratch training performs similarly on Open Benchmark but substantially better on in-house and long context tasks, highlighting pretraining value while showing CT conversion preserves and improves capabilities. Compression ratios stay at $R = 1.5$ throughout CT.

During PT (0-700B tokens), evaluation compression reaches $R = 1.81$ (12.5% FLOPs reduction), revealing distribution mismatch between PT and evaluation data. CT and evaluation data align better, both representing higher quality.

Table 6 shows post-SFT results. ConceptMoE-top15 gains +0.4 points, while ConceptMoE-top11-loop8 achieves +5.5 overall, with strong improvements in reasoning (+8.3), math (+12.2), and code (+6.4). Layer looping contributes, but crucially without increasing FLOPs. From-scratch training adds another +0.9 points. All configurations maintain $R = 1.5$ when evaluation.

*Table 6.* CT results on Open Benchmark. Comp. Eval.: Comprehensive Evaluation; FS: From Scratch.

| Category | MoE | Concept MoE-top15 | Concept MoE-top11 -loop8 | Concept MoE-top11 -loop8(FS) |
|---|---|---|---|---|
| Overall | 40.9 | 41.3 | 46.4 | **47.3** |
| Comp. Eval. | 49.4 | 50.5 | **54.4** | 53.9 |
| Reasoning | 30.3 | 28.4 | **38.6** | 38.5 |
| Math | 38.1 | 39.7 | 50.3 | **52.8** |
| Code | 20.0 | 20.9 | 26.4 | **30.1** |
| Instruction | **54.7** | 52.9 | **55.1** | 54.6 |
| Knowledge | **28.2** | 26.2 | 27.5 | 27.3 |
| Multilingual | 71.7 | 75.3 | 75.3 | **80.3** |

### 4.4. Significant inference speedup

We evaluated the inference latency of ConceptMoE on Hopper GPUs. Using MoE-A10B-300B as the baseline, we assessed 5 ConceptMoE configurations, ranging from efficiency-oriented to quality-oriented setups. Our results show that quality-oriented ConceptMoE achieves **comparable speed** to MoE on short sequences despite **doubling the number of layers**, while maintaining increasing speedup advantages on long sequences. Efficiency-oriented ConceptMoE delivers speedups of **32.1%** to **117.1%** in decoding and **24.7%** to **175.1%** in prefill.

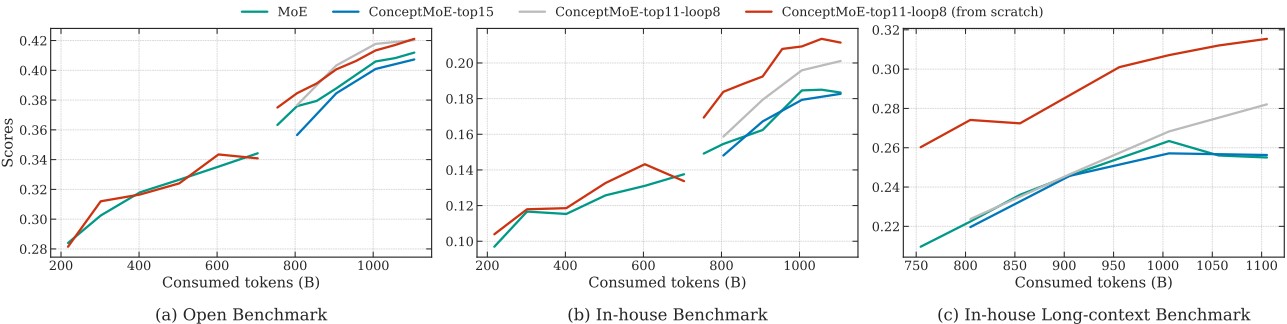

*Figure 4.* Evaluation metrics across three benchmarks during training.

Specifically, we configure models in the form ConceptMoE-$x$L-top$y$-R$z$, where $x$ represents the layer multiplication ratio, $y$ denotes the number of activated experts (baseline is 8), and $z$ indicates the compression ratio. For quality-oriented configurations, we use ConceptMoE-2L-top8-R2. For efficiency-oriented setups, we employ ConceptMoE-1L-top24-R2 and ConceptMoE-1L-top16-R1.5. For balanced configurations, we adopt ConceptMoE-1.5L-top13-R2 and ConceptMoE-1.25L-top11-R1.5. The $R = 1.5$ configurations are similar to the models discussed in Section 4.3.

Figure 5 shows the speedup ratio of ConceptMoE over MoE. For the prefill stage, we evaluate speedup variations across input sequences ranging from 4K to 1024K. For the decoding stage, we assess speedup changes for kv cache lengths from 4K to 64K at batch size 256. Figure 6 presents the actual latency. We observe that even when doubling the number of layers at $R = 2$, ConceptMoE still achieves substantial speedup, this stems from the quadratic reduction in attention map computation and the linear reduction in kv cache. When keeping the layer count unchanged, efficiency-oriented configurations achieve prefill speedups up to **175%**, which continue to increase with longer sequences, and decoding speedups up to **117.1%**. At $R = 1.5$, we also observe prefill speedups up to **43.6%** and decoding speedups up to **53.3%**. Section 4.3 validates the feasibility of integrating ConceptMoE in CT at $R = 1.5$, which allows us to directly convert existing MoE models to ConceptMoE losslessly while obtaining significant inference gains.

### 4.5. Structure ablation

We conduct extensive ablation studies to validate design choices (detailed results in Appendix B). Key findings include: (1) **Auxiliary loss weight**: $\lambda = 0.03$ balances compression ratio control and training loss, following H-Net (Hwang et al., 2025). (2) **Chunking strategy**: Dynamic chunking outperforms fixed-length merging by 2.2 points, with 0.004 lower training loss, validating adaptive boundary identification. (3) **Router design**: Cosine similarity-based router (36.4 average score) significantly outperforms linear

router (34.4), despite the latter achieving lower training loss, indicating better generalization through explicit semantic modeling. (4) **Joint decoding**: Removing joint decoding degrades downstream performance by 1.3 points despite lower training loss, confirming its role as implicit regularization. (5) **Boundary noise**: Applying Bernoulli noise with $\tau = 4$ improves robustness, gaining +1.4 points over baseline. (6) **Compression ratio**: $R = 2$ improves performance while $R = 4$ degrades substantially (47.7 vs 50.8), indicating optimal compression should match dataset redundancy rather than maximize compression unconditionally.

## 5. Conclusion

We introduce ConceptMoE, a framework that dynamically merges semantically similar tokens into concepts through learnable chunking. By operating within the MoE architecture, we enable fair comparison under controlled conditions: reallocating saved computation to match baseline FLOPs and total parameters isolates genuine architectural benefits. Experiments demonstrate consistent improvements across language pretraining (+0.9 points), vision-language training(+0.6 points), long context (+2.3 points), and continual training conversion (+5.5 points with layer looping).

Beyond performance gains, ConceptMoE reduces attention computation by up to $R^2\times$ and KV cache by $R\times$, achieving prefill speedups up to 175% and decoding speedups up to 117% at $R = 2$. The minimal architectural changes enable straightforward integration into existing systems. ConceptMoE demonstrates that adaptive concept-level processing fundamentally improves both the effectiveness and efficiency of LLMs, opening new directions for compute allocation in sparse architectures.

## Impact Statement

This paper presents work whose goal is to advance the field of Machine Learning. ConceptMoE introduces adaptive concept-level processing to improve both computational efficiency and performance of large language models.

The primary societal impact is environmental: by reducing computational costs through adaptive token compression (achieving prefill speedups up to 175% and decoding speedups up to 117%), our work potentially lowers energy consumption and carbon emissions associated with training and deploying large language models. The efficiency gains also improve accessibility, enabling researchers and practitioners with limited computational resources to work with advanced language models.

As with any advancement in language model capabilities and efficiency, ConceptMoE could potentially lower barriers to deploying powerful language models. We encourage practitioners to conduct domain-specific evaluations to ensure the adaptive compression strategy does not disproportionately affect particular content types or user groups, and to verify that compression patterns do not amplify existing biases in training data.

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

# A. Inference latency

We present inference latency for prefill and decoding with 300B total parameters and 10B activated parameters. Figure 5 shows the speedup of ConceptMoE over MoE across different configurations, while Figure 6 presents the actual latency measurements.

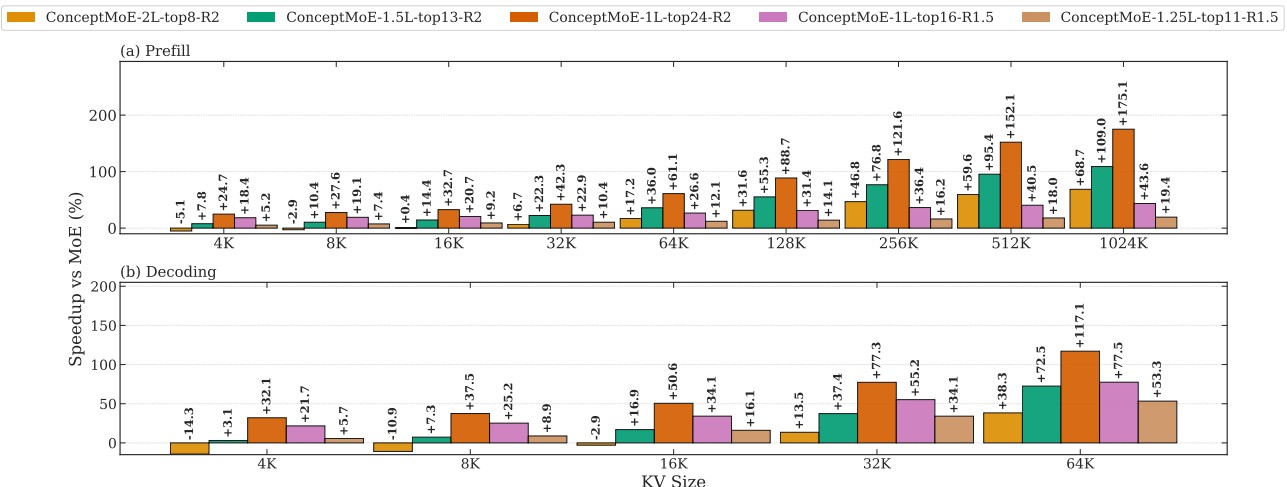

*Figure 5.* Inference latency speedup over MoE for prefill and decoding. The prefill plot uses sequence length on the x axis, and the decoding plot uses KV cache length on the x axis with batch size 256. The y axis reports speedup relative to MoE in percent. ConceptMoE-$x$L-top$y$-R$z$ matches MoE in FLOPs and total parameters, where $x$ is the layer multiplier, $y$ is the number of activated experts per MoE block with baseline 8 and increased to match FLOPs, and $z$ is the compression ratio.

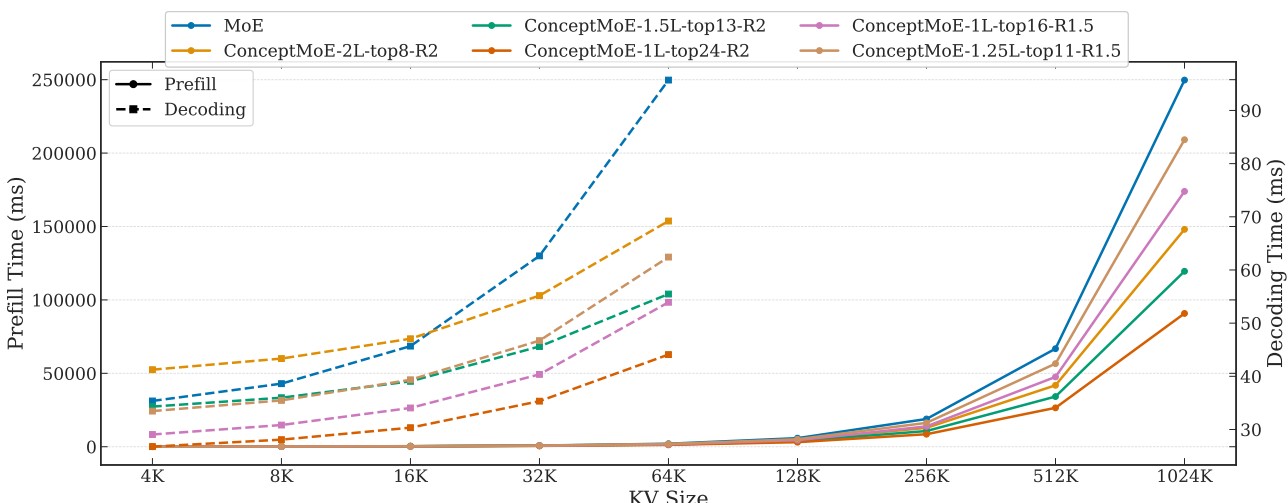

*Figure 6.* Inference latency for prefill and decoding. The prefill plot uses sequence length on the x axis, and the decoding plot uses KV cache length on the x axis with batch size 256. The y axis reports the measured end to end inference latency on Hopper GPUs. ConceptMoE-$x$L-top$y$-R$z$ matches MoE in FLOPs and total parameters, where $x$ is the layer multiplier, $y$ is the number of activated experts per MoE block with baseline 8 and increased to match FLOPs, and $z$ is the compression ratio.

# B. Structure ablation

### B.1. Auxiliary loss weight

We ablate the auxiliary loss weight $\lambda$ on ConceptMoE-A0.5B-12B trained for 243B tokens with target compression ratio $R = 2$. Following H-Net(Hwang et al., 2025)'s auxiliary loss weight of 0.03, we evaluate $\lambda \in \{0.03, 0.1, 0.5, 1.0\}$ and examine their effects on training loss and achieved compression ratio.

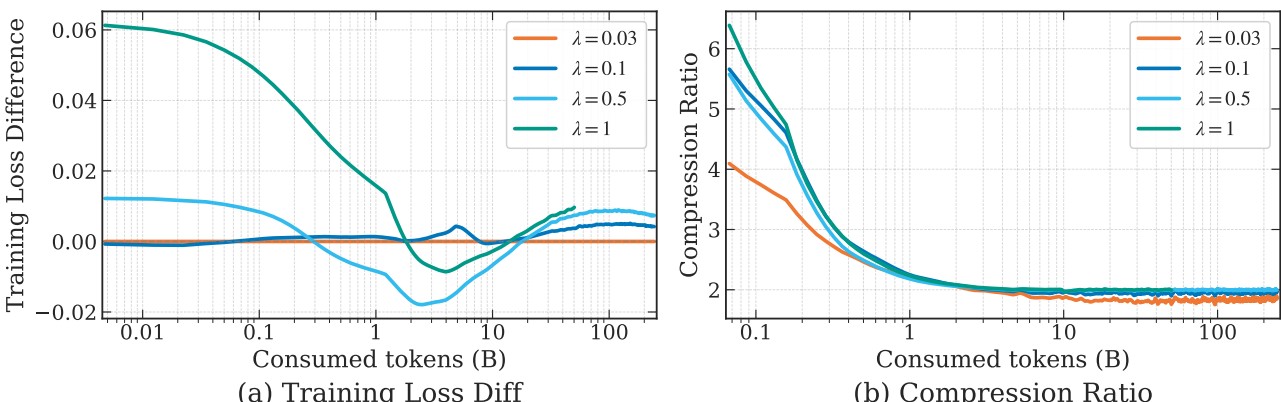

*Figure 7.* Impact of auxiliary loss weight $\lambda$ on training loss and compression ratio.

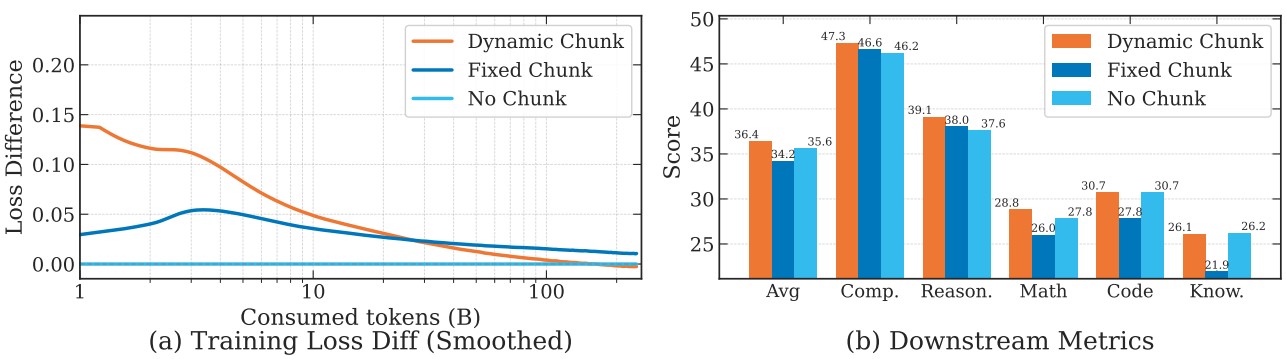

*Figure 8.* Comparison of chunking strategies. (a) Training loss difference relative to No Chunk baseline (i.e. MoE) during pretraining. Dynamic Chunk consistently achieves lower loss than Fixed Chunk. (b) Downstream benchmark scores after training. Comp.: Comprehensive Evaluation, Reason.: Reasoning, Know.: Knowledge.

Figure 7 presents the results. As $\lambda$ increases, training loss degrades while all configurations achieve compression ratios close to 2. Based on these findings, we set $\lambda = 0.03$ for all other experiments.

### B.2. Chunking strategy

We compare two chunking strategies on ConceptMoE-A0.5B-12B trained for 243B tokens with target compression ratio $R = 2$: (1) Dynamic Chunk: our learnable adaptive chunking based on token similarity, (2) Fixed Chunk: merging every consecutive $R$ tokens into one concept. We use No Chunk(MoE-A0.5B-12B) as our baseline.

Figure 8(a) shows training dynamics. Dynamic Chunk maintains a consistent advantage throughout training and ultimately achieves 0.004 lower loss than No Chunk, demonstrating that adaptive compression with proper compute reallocation improves optimization. In contrast, Fixed Chunk degrades by 0.01 relative to No Chunk, indicating that uniform merging disrupts the model's learning dynamics. Downstream evaluation (Figure 8(b)) confirms this pattern: Dynamic Chunk achieves the highest average score (36.4), outperforming No Chunk (35.6) and Fixed Chunk (34.2). These results validate that adaptive boundary identification preserves semantic coherence while enabling efficient computation.

### B.3. Router design

Section 3.2 introduces our cosine similarity-based score for identifying chunk boundaries. A natural alternative is to directly predict boundary scores using a linear layer, analogous to MoE routers. We compare these two designs: cosine router (our approach) and linear router, on ConceptMoE-A0.5B-12B trained for 243B tokens with target compression ratio $R = 2$.

Figure 9(a) shows that the linear router achieves 0.003 lower training loss at convergence. However, Figure 9(b) reveals a substantial gap in downstream performance: the linear router scores 34.4 average, significantly underperforming the cosine

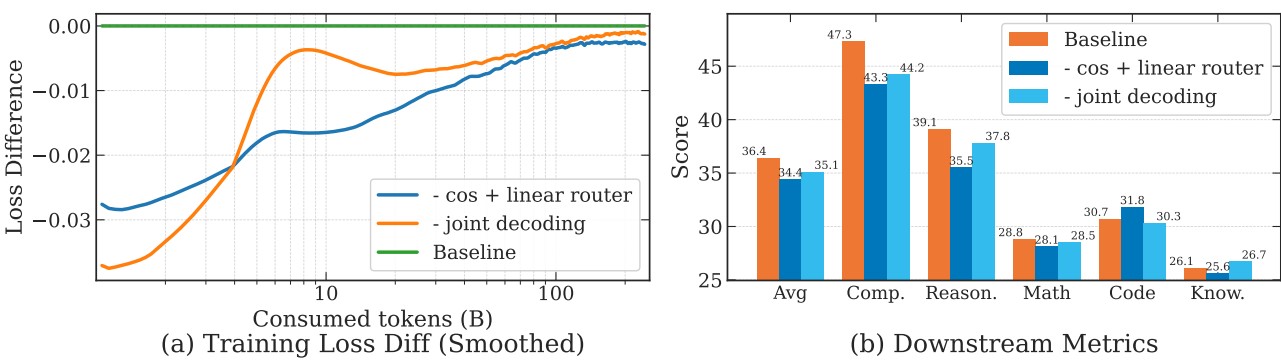

*Figure 9.* Impact of router type and joint decoding on training and downstream performance.

router at 36.4. This train-eval discrepancy suggests that the linear router overfits to training data patterns. In contrast, the cosine router's explicit modeling of inter-token similarity provides better generalization by capturing semantic relationships rather than memorizing dataset-specific boundary patterns.

### B.4. Joint decoding ablation

Section 3.4 introduces joint decoding of concepts and tokens in the decoder's final layers, adding negligible computation. We ablate this component on ConceptMoE-A0.5B-12B trained for 243B tokens with target compression ratio $R = 2$.

Figure 9(a) shows that removing joint decoding yields 0.002 lower training loss at convergence. However, Figure 9(b) reveals substantial downstream degradation: the model without joint decoding scores 35.1 average compared to 36.4 with joint decoding. This pattern mirrors the router ablation: improvements in training loss do not guarantee better generalization.

We hypothesize that joint decoding acts as implicit regularization during training. By forcing the decoder to explicitly attend to concept information through additional QKV projections, the model learns more robust representations that transfer better to downstream tasks. Without joint decoding, the decoder may overfit to residual token-level patterns in the training data while underutilizing the semantic information encoded in concepts. This validates that leveraging concept representations throughout the decoding process is essential for realizing the full benefits of adaptive compression.

### B.5. Boundary Noise for Robustness

We observe a discrepancy between training and evaluation compression ratios: the model achieves lower compression during training than during evaluation. Analysis of the boundary probability distribution reveals that a substantial portion of probabilities cluster around 0.5. During evaluation, these borderline cases can easily flip, causing unintended compression ratio drift.

Theoretically, for target compression ratio $R$, the mean probability should satisfy $\mathbb{E}[p_n] \approx 1/R$. However, without noise, we observe significantly lower probability means, indicating that many boundaries hover near the 0.5 threshold. During evaluation, slight perturbations or distribution shifts push these marginal probabilities over the boundary, leading to higher compression than intended.

To improve robustness, we introduce noise during training to simulate evaluation conditions. We evaluate Bernoulli noise with $\tau \in \{4, 6\}$ for $p_n^{sharp}$ and Gaussian noise with $\sigma = 0.1$. All models train with target $R = 1.5$.

Figure 10 shows training dynamics. Bernoulli noise strategies yield higher training loss at convergence, with smaller $\tau$ producing larger degradation, confirming that noise perturbs chunk boundaries. Gaussian noise maintains comparable training loss. Critically, noise regularization normalizes the probability mean closer to $1/R$ (e.g. 0.667), stabilizing evaluation compression ratios.

Table 7 demonstrates that noise improves downstream performance despite higher training loss. ConceptMoE with $\tau = 4$ achieves 30.3 average score (+1.4 over baseline), with stronger Bernoulli noise proving most effective. This validates that boundary noise strengthens chunk module robustness by preventing probability collapse around 0.5, ensuring consistent compression behavior between training and evaluation. To avoid excessive impact on training loss, we use Bernoulli noise

with $\tau = 6$ in all other experiments.

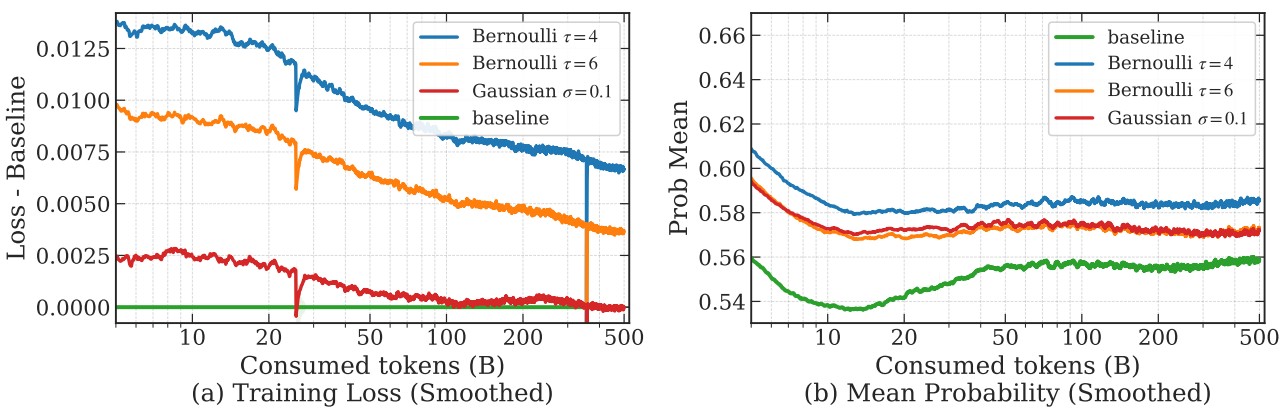

*Figure 10.* Training loss diff and mean probability of chunk module for different noise strategies. We do smoothing and remove some spike data for better visualization, which doesn't affect the conclusion.

*Table 7.* Model performance comparison on OpenBench for different noise strategies. Comp.: Comprehensive Evaluation, Spec.: Specialized Knowledge, Know.: Knowledge.. Comp.: Comprehensive Evaluation, Reason.: Reasoning, Know.: Knowledge.

| Category | ConceptMoE | ConceptMoE-Gaussian | ConceptMoE-Tau4 | ConceptMoE-Tau6 |
|---|---|---|---|---|
| All | 28.9 | 29.6 | **30.3** | 30.0 |
| Comp. Eval. | 38.9 | 40.0 | 40.1 | **40.3** |
| Reasoning | 22.9 | 23.5 | **24.7** | 24.0 |
| Math | 9.1 | 8.7 | 9.4 | **9.5** |
| Code | 36.0 | 37.5 | **38.8** | 38.7 |
| Spec. | 25.0 | **28.4** | 27.6 | 26.1 |
| Know. | 30.2 | 28.6 | **30.3** | 29.6 |

## B.6. Target compression ratio

We investigate the impact of compression ratio by training models with $R = 2$ and $R = 4$, using MoE-A1B-24B as baseline with compute reallocation strategy 3 (increasing $C_{attn}$ and $C_{moe}$ from Section 3.5) over 559B tokens.

Figure 11 reveals that higher compression does not guarantee better performance. While $R = 2$ achieves comparable training loss to the baseline and improves downstream scores to 50.8 average, $R = 4$ shows substantial degradation in both training loss (0.013 gap at convergence) and downstream performance (47.7 average). The gap is particularly pronounced on reasoning (51.3 vs 56.8) and math (46.3 vs 50.0), suggesting that aggressive compression disrupts complex reasoning patterns.

We hypothesize that each dataset has an optimal compression ratio determined by its semantic redundancy distribution. At $R = 4$, the model is forced to merge tokens with significant semantic differences, losing critical information for downstream tasks. The auxiliary loss successfully constrains the compression ratio during training, but the resulting $4\times$ compression fundamentally exceeds the natural redundancy level in the data. This indicates that compression ratio should be calibrated to dataset characteristics rather than maximized unconditionally. For typical pretraining corpora, $R = 1.5$ to $R = 2$ appears to strike an effective balance between efficiency and information preservation.

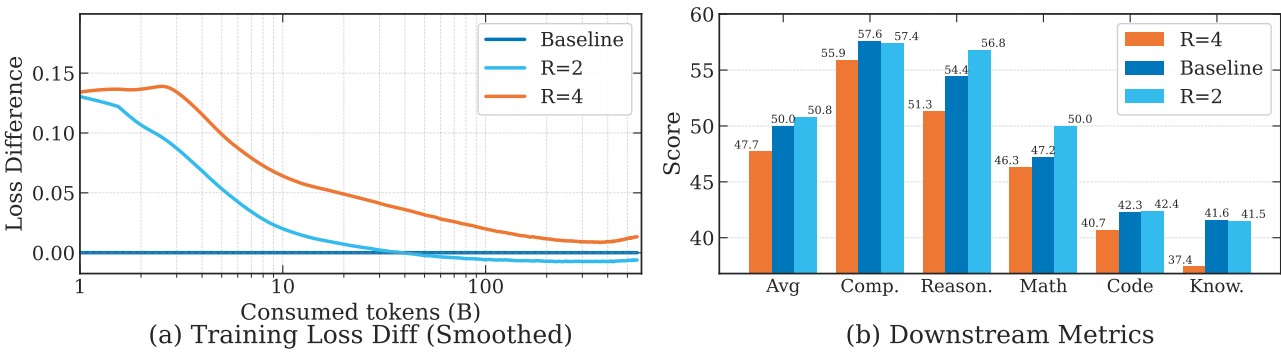

*Figure 11.* Impact of target compression ratio on training and downstream performance. (a) Training loss difference relative to baseline. $R = 2$ converges close to baseline, while $R = 4$ shows persistent degradation. (b) Downstream benchmark scores. $R = 2$ outperforms the baseline across most metrics, while $R = 4$ underperforms substantially, particularly on reasoning and math tasks. Results indicate that excessive compression (e.g., $R = 4$) degrades performance despite successful compression ratio control.

## C. Evaluation benchmark

We provide the evaluation datasets included in each category of OpenBench. Table 8 details the relatively easy evaluation sets used for models with 12B and 24B total parameters. Table 9 includes both easy and hard evaluation sets used for models with 60B and 90B total parameters.

*Table 8.* Open benchmarks Easy across different domains. Citations: MMLU(Hendrycks et al., 2021b;a), BBH(Suzgun et al., 2022), MATH(Hendrycks et al., 2021c), HumanEval(Chen et al., 2021), TriviaQA(Joshi et al., 2017), C-Eval(Huang et al., 2023), DROP(Dua et al., 2019), MBPP+(Austin et al., 2021), ChineseSimpleQA(He et al., 2024b), MMLU-Pro(Wang et al., 2024), McEval(Chai et al., 2024), AGIEval(Zhong et al., 2023).

| Comprehensive Evaluation | Reasoning | Math | Code | Knowledge |
|---|---|---|---|---|
| MMLU | BBH | MATH | HumanEval | TriviaQA |
| C-Eval | DROP | | MBPP+ | ChineseSimpleQA |
| MMLU-Pro | | | McEval | |
| AGIEval | | | | |

*Table 9.* Open benchmarks across different domains. Citations: MMLU (Hendrycks et al., 2021b;a), BBH (Suzgun et al., 2022), MATH (Hendrycks et al., 2021c), HumanEval (Chen et al., 2021), TriviaQA (Joshi et al., 2017), GPQA (Rein et al., 2024), C-Eval (Huang et al., 2023), DROP (Dua et al., 2019), MBPP+ (Austin et al., 2021), SimpleQA (Wei et al., 2024), MMLU-Pro (Wang et al., 2024), ARC_AGI (Chollet, 2019), LiveCodeBench (Jain et al., 2024), ChineseSimpleQA (He et al., 2024b), AGIEval (Zhong et al., 2023), ProcBench (Fujisawa et al., 2024), HARP (Yue et al., 2024), McEval (Chai et al., 2024), HLE (Phan et al., 2025), ZebraLogic (Lin et al., 2024), Omni-MATH (Gao et al., 2024), SuperGPQA (Team et al., 2025), KOR-Bench (Ma et al., 2024), OlympiadBench (He et al., 2024a), LiveBench (White et al., 2025).

| Comprehensive Evaluation | Reasoning | Math | Code | Knowledge | Specialized Knowledge |
|---|---|---|---|---|---|
| MMLU | BBH | MATH | HumanEval | TriviaQA | GPQA |
| C-Eval | DROP | AIME2024 | MBPP+ | SimpleQA | |
| MMLU-Pro | ARC_AGI | AIME2025 | LiveCodeBench | ChineseSimpleQA | |
| AGIEval | ProcBench | HARP | McEval | | |
| HLE | ZebraLogic | Omni-MATH | | | |
| SuperGPQA | KOR-Bench | OlympiadBench | | | |
| LiveBench | | | | | |

# D. Code

Below we present the forward function in PyTorch format. In this example, the encoder has 2 layers, the concept model has 23 layers, and the decoder has 2 layers. The hidden size is 2048. The input consists of 1024 tokens, which are compressed to 701 tokens before entering the concept model.

```python
import torch
import torch.nn.functional as F
from torch import nn

class ChunkModule(nn.Module):
    def __init__(self, config):
        super().__init__()
        self.q_proj_layer = nn.Linear(config.hidden_size, config.hidden_size, bias=False)
        self.k_proj_layer = nn.Linear(config.hidden_size, config.hidden_size, bias=False)

    def forward(self, hidden_states):
        cos_sim = torch.einsum(
                "l d, l d -> l",
                F.normalize(self.q_proj_layer(hidden_states[:, :-1]), dim=-1),
                F.normalize(self.k_proj_layer(hidden_states[:, 1:]), dim=-1),
                )          # shape [1023,]
        boundary_prob = torch.clamp(((1 - cos_sim) / 2), min=0.0, max=1.0)   # shape [1023
                                            ,]
        # Force boundary probability of the first element to 1.0
        PAD_PROB = 1.0
        boundary_prob = F.pad(boundary_prob, (1, 0), "constant", PAD_PROB) # shape [1024,]

        selected_idx = torch.zeros_like(boundary_prob, dtype=torch.long)
        boundary_mask = boundary_prob >= 0.5
        selected_idx[..., boundary_mask] = 1
        boundary_prob = torch.stack(((1 - boundary_prob), boundary_prob), dim=-1)

        selected_probs = boundary_prob.gather(
                dim=-1, index=selected_idx.unsqueeze(-1)
            )   # (shape hidden_states.shape[:-1], 1)
        return boundary_prob, boundary_mask, selected_probs

class DechunkModule(nn.Module):
    def __init__(self, config):
        super().__init__()

    def forward(self, concept, boundary_prob, boundary_mask):
```

```python
        concept_prob = boundary_prob[boundary_mask]      # shape [701,]
        concept_merge = torch.zeros_like(concept)

        # For ease of understanding, this is written in for-loop form. In practice, it can
                                          be accelerated through parallel scan
                                          .
        concept_merge[0] = concept[0]                    # shape [701, 2048]
        for i in range(1, concept.shape[0]):
            concept_merge[i] = concept_merge[i-1]*(1-concept_prob[i]) + concept[i] *
                                          concept_prob[i]

        plug_back_idx = boundary_mask.cumsum(dim=0) - 1
        concept_merge = torch.gather(
                concept_merge, dim=0, index=plug_back_idx.expand(-1, 2048)
            )                # concept_merge shape [701,2048] -> [1024,2048]
        return concept_merge

class STE(torch.autograd.Function):
    @staticmethod
    def forward(ctx, x):
        return torch.ones_like(x)
    @staticmethod
    def backward(ctx, grad_output):
        grad_x = grad_output
        return grad_x
def ste_func(x):
    return STE.apply(x)

class ConceptMoE(nn.Module):
    def __init__(self, config):
        super().__init__()
        self.encoder = nn.ModuleList(transformer_layer(layer_id=i) for i in range(2))
        self.concept_model = nn.ModuleList(transformer_layer(layer_id=i) for i in range(2,
                                          25))
        self.decoder = nn.ModuleList(transformer_layer(layer_id=i) for i in range(25,27))

        self.lm_head = nn.Linear(config.hidden_size, config.vocab_size, bias=False)
        self.embedding = nn.Embedding(config.vocab_size, config.hidden_size)

        self.chunk_module = ChunkModule(config)
        self.dechunk_module = DechunkModule(config)

    def forward(self, input_ids):
        # encoder
        hidden_state = self.embedding(input_ids)        # shape [1024, 2048]
        hidden_state = self.encoder(hidden_state)

        # chunk
        boundary_prob, boundary_mask, selected_probs = self.chunk_module(hidden_state)

        # main network
        concept = hidden_state[boundary_mask]            # shape [701, 2048]
        concept = self.concept_model(concept)

        # dechunk
        concept_merge = self.dechunk_module(concept, boundary_prob, boundary_mask)

        # decoder
        hidden_state = hidden_state + concept_merge * ste_func(selected_probs)
        hidden_state = self.decoder(hidden_state  concept_merge)   # joint decoding

        logits = self.lm_head(hidden_state)
        return logits
```

