# OpenReview forum: "ConceptMoE: Adaptive Token-to-Concept Compression for Implicit Compute Allocation"
_ICML.cc/2026/Conference — ICML 2026 regular_

### Official Review · Reviewer_EPim · 2026-03-01

**Soundness:** 3
**Presentation:** 3
**Significance:** 3
**Originality:** 2
**Overall Recommendation:** 4
**Confidence:** 3

**Summary:**

ConceptMoE is an innovative Large Language Model (LLM) architecture designed to address the efficiency bottleneck in traditional models where equal computational power is allocated to every token. It utilizes a learnable chunking module to dynamically merge tokens into variable-length "concepts" based on semantic similarity, which are then processed by a Mixture-of-Experts (MoE) backbone for deep reasoning. By redistributing resources under the same parameter count and computational budget, this work demonstrates significant performance advantages in pre-training, long-context, and multi-modal tasks while substantially increasing inference speed.

**Compliance With Llm Reviewing Policy:**

Affirmed.

**Final Justification:**

I maintained my current positive score.

**Key Questions For Authors:**

1. Is there room to further increase $R$? Given that $R < 2$ at present, the engineering trade-offs required to adapt to this architecture seem quite significant.

Minor Questions:

2. Why does the decrease in training loss not correspond to an improvement in performance? You used a relatively large training dataset that should theoretically cover various domains, yet the effectiveness on downstream tasks remains uncertain.

3. Could you provide a comparison of the differences in compression ratio $R$ across tasks of varying difficulty?

4. What is the maximum chunk size observed during experiments?

**Limitations:**

yes

**Strengths And Weaknesses:**

Strength:
1. The experiments are extensive and the comparisons are fair, ensuring that FLOPs are aligned.
2. A wide range of benchmarks were tested, and large-scale model parameters were used to verify effectiveness.
3. It proves the potential prospects of the "concept model" approach.

Weakness:
1. Table 5 exceeds the page margins.
2. The core underlying designs are largely derived from previous work; this paper primarily focuses on extending the method to MoE models.
3. Considering the potential engineering adaptations required, the current compression ratio $R$ is not particularly high.

---

> ### Author Rebuttal · Authors · 2026-03-30
>
> We thank the reviewer for the positive assessment of the experiments, the fairness of the matched-FLOPs setting, and the promise of the concept-model direction.
>
> **(1) Is there room to further increase the compression ratio R, given the engineering trade-offs?**
>
> Yes. Our current results suggest that R=2 is already a strong practical sweet spot, while still leaving room for higher compression.
>
> To better answer this question, we additionally trained a larger-scale ConceptMoE model (700B from-scratch pretraining + 32k CT on 400B + 128k CT on 50B + SFT) with R=2. We reallocated the saved compute while keeping the total parameter count and per-token FLOPs matched to the baseline. Under this controlled setting, ConceptMoE improves the overall score by +5.1. This shows that R=2 is not only viable, but highly effective at scale. From an engineering perspective, R=2 is already attractive, with estimated speedups of up to 2.52x in prefill and 2.06x in decoding.
>
> At the same time, R=4 is more challenging, suggesting that higher compression introduces a harder optimization / information-preservation problem. We will clarify in the revision that R=2 currently appears to be the best practical operating point, while higher-R settings remain promising future work.
>
>
> | Model | Overall |
> |---|---:|
> | MoE | 40.9 |
> | ConceptMoE-top15 | 41.3 |
> | ConceptMoE-wide (R=2) | 45.9 (+5.1) |
>
> **(2) Why does lower training loss not always translate into proportional downstream gains?**
>
> This is a good question. In Sec. 4.3, we observe that after PT, downstream performance is roughly on par with the baseline, while clear gains appear after CT and become largest after SFT. We believe there are two main reasons.
>
> First, after PT the average realized compression ratio on downstream evaluation is about 1.8, while after CT it becomes about 1.5, which is closer to the intended target(1.5). This means the PT-stage downstream evaluation actually uses fewer FLOPs.
>
> Second, we believe this reflects a data-distribution shift. Downstream evaluation data are typically higher quality and better matched to the CT data than to the broad PT corpus. As a result, the compression behavior learned after CT transfers better to downstream tasks, whereas PT improvements in training loss do not necessarily yield proportional downstream gains.
>
> **(3) Could you compare compression ratios across tasks of different difficulty?**
>
> Yes. We measured realized compression ratios on downstream domains under a model trained and evaluated with average target ratio R=2:
> | Comprehensive set | Reasoning | Math | Code | Instruction | Knowledge | Multilingual |
> |---:|---:|---:|---:|---:|---:|---:|
> | 1.96 | 2.56 | 1.96 | 3.70 | 1.64 | 1.33 | 1.64 |
>
> These results suggest that compression ratio is not determined simply by task difficulty. Reasoning and code show relatively high compression ratios, while knowledge has the lowest. We believe the learned boundary mechanism responds more to local redundancy and structural regularity: chain-of-thought reasoning and code contain recurring patterns that are easier to merge into larger concepts, whereas knowledge tasks are less repetitive and thus preserve finer-grained boundaries. See also our **response to Reviewer qpui (3)** for qualitative chunk-boundary examples.
>
> **(4) What is the maximum chunk size observed in experiments?**
>
> We tracked chunk-size statistics during training. For the model trained with target compression ratio R=2, the maximum observed chunk size is 163; chunk-size p95 is 4.6, p90 is 3.3, and p50 is 1. About 60% of chunks have size 1.
>
> For the model trained with target compression ratio R=4, the maximum observed chunk size is 229; chunk-size p95 is 7.9, p90 is 5.9, and p50 is 2.6. About 25% of chunks have size 1.
>
> These statistics suggest that although the model occasionally forms very large chunks, most chunks remain relatively small; the large maximum chunk sizes are rare outliers rather than typical behavior. As expected, moving from R=2 to R=4 shifts the distribution toward larger chunks.
>
> **(5) Table 5 exceeds the page margins.**
>
> Thank you for catching this. We will fix the formatting issue in the camera-ready version.
>
> **(6) “The core designs are largely derived from prior work; the paper mainly extends them to MoE.”**
>
> We appreciate this point. Our claim is not that every module is individually unprecedented, but that we introduce an MoE-compatible adaptive token-to-concept mechanism that enables implicit compute reallocation under matched parameter / FLOPs budgets. The novelty lies in integrating adaptive boundary learning, concept-level MoE processing, dechunking, and joint decoding into a controlled compute-allocation framework. We will make this claim more explicit and carefully scoped in the revision.
>
> We appreciate the reviewer’s thoughtful questions and hope these clarifications address the main concerns. If the reviewer finds them helpful, we would be grateful for reconsideration of the evaluation.

---

> > ### Author Rebuttal · Reviewer_EPim · 2026-04-01
> >
> > Thanks for the rebuttal. I will increase my significance score.

---

### Official Review · Reviewer_KfMD · 2026-03-10

**Soundness:** 2
**Presentation:** 1
**Significance:** 2
**Originality:** 2
**Overall Recommendation:** 3
**Confidence:** 3

**Summary:**

This paper proposes ConceptMoE, a mixture-of-experts architecture that performs adaptive token-to-concept compression via a learnable chunking module that merges adjacent, semantically similar tokens before applying the compute-heavy “concept model”. It introduces a simple cosine-similarity-based boundary detector with an auxiliary ratio loss, an EMA-based dechunking mechanism, and “joint decoding” that fuses concept representations into the final layers’QKV projections.

**Compliance With Llm Reviewing Policy:**

Affirmed.

**Final Justification:**

I appreciate the authors' rebuttal clarification on the contributions/challenges, so I increase my score to 3. There are some other remaining issues to fix after rebuttal: e.g., MoE introduction, emprical evidence on the limitations of existing methods, adding other baseline methods, and presentations.

**Key Questions For Authors:**

1. The problem is not sufficiently elaborated. While the paper mentions “MoE” from time to time, a formal introduction of “MoE” has never been given. Nor does the paper present an informative study on how necessary/significant it is to conduct similarity-based chunk compression.
2. This paper reads like a plain technical report. It is hard to judge the novelty of this paper. The authors fail to present the design objectives (i.e., what are the desired characteristics of a good solution). Nor did the authors formally discuss the challenges when enforcing the key insight. Besides, in each module, the authors did not discuss the candidate solution space. With a mess of technical details, it is hard to judge the superiority of the proposed solutions.
3. In evaluation there is a lack of formal baselines. The only baseline is MoE, which as the default practice, is far from sufficient. Meanwhile, the ablation and sensitivity study (at least for the most representative evaluation results in each category) shall be placed into the main paper body given their significance.
4. There are many presentation flaws, rendering this paper not ready for publication in this current shape. For example, “separatelywe”, “limitedda”, “allocationthe”, etc. The presentation could be substantially improved.

**Limitations:**

Yes.

**Strengths And Weaknesses:**

Strengths：
1. It is a meaningful research topic to explore the concept-level token representation for higher inference efficiency.
2. The idea of adaptively identifying semantic units and processing at concept level rather than processing at a fixed token granularity makes sense.
3. Evaluations are conducted on diverse task categories.

Weaknesses：
1. The problem is not sufficiently elaborated.
2. This paper reads like a plain technical report. It is hard to judge the novelty of this paper.
3. In evaluation there is a lack of formal baselines.
4. There are many presentation flaws, rendering this paper not ready for publication in this current shape.

---

> ### Author Rebuttal · Authors · 2026-03-30
>
> We thank the reviewer for the careful reading and candid feedback.
>
> **(1) The problem is not sufficiently elaborated, and MoE is not formally introduced.**
>
> The paper studies a specific problem: although MoE provides conditional computation across experts, it still processes sequences at uniform token granularity, so semantically light or predictable spans may still receive unnecessarily expensive computation. ConceptMoE addresses this by adaptively merging locally coherent token spans into concept-level units, so that the saved sequence-level compute can be reallocated more effectively inside the MoE backbone.
>
> Under this formulation, MoE is not incidental background but the core setting of the paper. Our goal is not generic token compression in isolation, but concept-level compression for better compute allocation in sparse MoE systems. The current submission already provides supporting evidence: dynamic chunking substantially outperforms fixed chunking, the auxiliary ratio loss prevents degenerate compression, and the gains persist across language, multimodal, and continual-training settings. We agree, however, that the paper should introduce standard MoE more explicitly at the beginning. We also refer the reviewer to our response to **Reviewer qpui (3)**, where we further analyze what the learned dynamic boundaries capture.
>
> **(2) The paper reads like a plain technical report, and the novelty / design objectives / challenge discussion are unclear.**
>
> We believe the core objectives and challenges are already present, but not surfaced clearly enough. The objectives are to: (i) adaptively identify compressible spans instead of allocating uniform token-level compute, (ii) preserve token-level outputs for autoregressive modeling, (iii) use compression to reallocate saved compute more effectively inside MoE, and (iv) remain compatible with existing MoE infrastructure.
>
> The corresponding challenges are reflected in the method design: compression can become degenerate without ratio control; learned boundaries must outperform fixed segmentation; concept-level computation must still support token-level prediction; and gains should come from better compute allocation rather than simply enlarging the model. This is why the paper includes the auxiliary ratio loss, dynamic-vs-fixed chunking, dechunking / joint decoding, and matched-budget evaluation.
>
> We also note that not every architectural part is a method-specific design choice. The encoder -> concept model -> decoder scaffold is standard in token-compression / concept-modeling methods. Our contribution is the method-specific design that makes this scaffold effective for MoE compute allocation, including adaptive boundaries, ratio-controlled compression, concept-level MoE processing, and joint decoding. We agree this distinction should be made more explicit.
>
> **(3) There is a lack of formal baselines.**
>
> Since the central question is whether adaptive token-to-concept compression improves compute allocation within MoE, the most appropriate primary baseline is a carefully matched MoE backbone. MoE is the practically relevant sparse architecture for the problem studied here.
>
> At the same time, the empirical support is broader than may have been apparent from the presentation. The paper compares across baselines with multiple MoE scales, multimodal MoE models, and continual-training conversion settings.
>
> Beyond the main paper, our other rebuttal experiments further show that the gain is not tied to a single setup: in **response to Reviewer Fv85** we report a no-extra-activated-expert variant with about 2x speedup that still outperforms a naively downsized MoE, and in **response to Reviewer EPim** we report a from-scratch R=2 model with +5.1 overall score and >2x inference speedup.
>
> We agree that the ablations and sensitivity studies are important. Their placement in the appendix was mainly due to page limits: the main paper prioritizes scaling, multimodal transfer, continual-training integration, and real-system speed measurements, while the appendix contains more detailed method-analysis experiments.
>
> **(4) Many presentation flaws, including typos and formatting issues.**
>
> We agree with this point and sincerely apologize for the presentation issues such as spacing and table formatting overflow. These issues hurt readability and are entirely our responsibility. We will carefully proofread and fix them.
>
> Overall, we hope this clarifies that the paper’s main contribution is an adaptive concept-level computation mechanism for MoE that improves compute allocation under controlled budgets, with consistent empirical support across multiple settings. We appreciate the reviewer’s feedback, and if these clarifications address the main concerns, we would be grateful for reconsideration of the evaluation.

---

> > ### Author Rebuttal · Reviewer_KfMD · 2026-04-04
> >
> > Thanks for your rebuttal. I appreciate your clarification on the contributions/challenges. I will increase my score to 3. Please fix other issues in the later version of this paper, e.g., MoE introduction (can your method also be applied for non-MoE models?), emprical evidence on the limitations of existing methods, other baseline methods (only two---MoE and ConceptMoE---are too weak), and presentations.

---

### Official Review · Reviewer_Fv85 · 2026-03-11

**Soundness:** 3
**Presentation:** 3
**Significance:** 3
**Originality:** 2
**Overall Recommendation:** 4
**Confidence:** 2

**Summary:**

The paper proposes ConceptMoE, a dynamic token compression framework. The method dynamically groups tokens into higher-level units, processes these compressed representations with MoE, and reallocates the saved compute toward stronger processing. The authors claim increases in accuracy and efficiency.

**Compliance With Llm Reviewing Policy:**

Affirmed.

**Final Justification:**

The paper introduces an interesting idea with potential relevance for improving efficiency in sequence modeling, and I find the direction promising. The approach appears technically sound within the presented setup, and the analysis provides useful initial insights. However, I have concerns regarding the strength of the novelty and empirical validation relative to existing compression or bottleneck approaches.

The authors’ rebuttal was helpful in clarifying several aspects, including the role of layer looping and activated parameters, additional ablations, and assumptions around FLOPs accounting. These clarifications increased my confidence in parts of the methodology. However, my core concerns remain only partially addressed. In particular, the lack of direct comparison to other token-/byte-level compression baselines makes it difficult to assess the true added value of the approach beyond the controlled setting. Additionally, while qualitative examples of learned chunking are informative, the paper still lacks more systematic, statistically grounded evidence that these boundaries align with meaningful semantic units.

Overall, while I see promise in the idea, I weigh the current limitations in empirical validation and support for the main claims against its strengths. The rebuttal partially improved my assessment but did not fully resolve these central concerns, which I believe would require a more substantial revision to address.

**Key Questions For Authors:**

1. The paper suggests that benefit comes from concept-level processing, but the actual architecture has additional complexity, changing hidden size, number of activated experts, and loop layers. Can the authors provide further ablations/analysis for each ingredient: layer looping changes only, expert-count changes only, etc.? (The chunking module itself contains a lot of performance-optimizing components too: auxiliary loss, random flip boundary, where ablations are properly provided) The work will benefit from a more structured analysis of why each structure of the complex model is necessary and what is the most significant contributor.

2. Although the model maintains an identical total parameter count and per-token FLOPs, you do vary the count of activated parameters. Could the authors report the model performance if no parameter is increased? Does the efficiency claim hold solidly without degrading accuracy?

3. What is the strongest novelty claim relative to related work like H-Net and DLCM, if we separate the fairness protocol from the architecture itself? Notice that the architecture is indeed different, but please defend the necessity of these changes more elaborately.

4. Do the learned chunk boundaries actually correspond to meaningful semantic units (AKA concepts)? Please provide statistically significant evidence to support this claim, or at least an illustration of the segmentation working semantically.

**Limitations:**

Yes

**Strengths And Weaknesses:**

Strength:

1. Strong and comprehensive experiments that clearly show ConceptMoE's empirical advantage, with diversified settings featuring different model scales.

2. Reasonable framing of alternative compute allocation strategy: compressing inputs as concepts and using the saved FLOP counts.

3. Dynamic token compression, as suggested, is an important technique with downstream advantages. If the claims made in this paper are generalizable and surpass other compression strategies (not sufficiently clear yet), the model would signal a strong contribution.

Weakness:
1. The main baseline provided, MoE, is helpful as a mechanistic insight in comparing the allocation strategy, but to make a convincing novel-not-incremental claim, there should be more comparisons with other token-level and byte-level chunking models and bottleneck/compression methods.

2. Since the model is complex (encoder/chunk/concept/dechunk/decoder modules, auxiliary loss, random boundary flipping, merging choices, and joint decoding), it's not clear which component of the model is its deciding factor for success, and not every auxiliary structure is justified.

3. FLOPs comparisons exclude attention map computation, suggesting a weaker-than-expected fairness claim.

---

> ### Author Rebuttal · Authors · 2026-03-30
>
> We thank the reviewer for the thoughtful comments and helpful questions.
>
> **From weaknesses**
>
> **(1) Comparison to other token-/byte-level chunking and compression methods.**
>
> We agree that broader baseline coverage would strengthen the paper. Our focus is to test whether adaptive token-to-concept compression improves compute allocation inside MoE under matched parameter/FLOPs budgets, so the primary baseline is a carefully controlled MoE model. The dynamic-vs-fixed chunking ablation already shows that the gain comes from learning where to compress rather than compressing uniformly. Please also see Key Question (3) below for our positioning relative to H-Net and DLCM.
>
> **(2) Which component is the deciding factor? Why are all components necessary?**
>
> We agree this can be presented more clearly. The paper already ablates the auxiliary ratio loss, dynamic vs. fixed chunking, router design, joint decoding, boundary noise, and compression ratio, showing that the gain is not due to a single heuristic. Please also see Key Questions (1) and (2) below on layer looping and activated-parameter changes.
>
> **(3) FLOPs accounting excluding attention-map computation.**
>
> We agree that attention-map FLOPs should be stated explicitly. However, omitting this term makes the comparison more conservative, not more favorable to ConceptMoE, because ConceptMoE always has lower attention-map FLOPs after sequence compression.
>
> **From key questions**
>
> **(1) Layer looping / expert count / activated-parameter changes.**
>
> Layer looping (R=1.5) is only used in continual-training conversion and is not required for the gains in pretraining or multimodal evaluations. To isolate this point, we additionally trained a from-scratch non-looping variant with R=2 (see response **Reviewer EPim (1)**). With minimal changes to the baseline, it still improves the overall score by +5.1, showing that the main gain comes from concept-level processing and compute reallocation rather than from layer looping.
>
> More broadly, choices such as hidden size and expert count are only used to ensure fair comparison under matched total parameters and per-token FLOPs. For a given baseline, a budget-matched ConceptMoE with slightly larger hidden size and fewer experts can performs better than the baseline.
>
> **(2) What if no activated parameter is increased? Does the efficiency claim still hold?**
>
> Yes. We additionally trained a variant without increasing the number of activated experts during continual training, denoted as ConceptMoE-top8. After 400B continual training followed by SFT, ConceptMoE-top8 is only 1.8 aggregate points below the MoE baseline while using about 25% lower average per-token FLOPs under the accounting used in the current submission. This implies about 1.33x additional efficiency headroom relative to the matched-budget setting, corresponding to expected speedups of up to 1.9x in prefill and 2.36x in decoding.
>
> A naively downsized MoE at a similar speed regime typically degrades much more in our experience (around 5 aggregate points). Thus, the efficiency claim remains valid even without increasing activated parameters: ConceptMoE can still trade a modest quality drop for substantial efficiency gains.
>
> | Model | Aggregate Score |
> |---|---:|
> | MoE-top8 | 40.1 |
> | ConceptMoE-top15 | 40.3 |
> | ConceptMoE-top8 | 38.3|
>
> **(3) What is the strongest novelty claim relative to H-Net and DLCM, if we separate the fairness protocol from the architecture itself? Why are these changes necessary?**
>
> Our claim is not that every low-level component is individually unprecedented. The main novelty is at the system level: ConceptMoE is an adaptive token-to-concept compression mechanism designed for MoE conditional computation, so that compute saved by sequence compression can be converted into stronger expert computation under controlled budgets.
>
> Relative to H-Net, our setting is token-level rather than byte-level, targeting modern tokenizer-based LLM/MoE pipelines. Relative to DLCM, the key difference is the compute-allocation mechanism: DLCM is developed in a dense setting and matches baseline per-token FLOPs by adding layers with more parameters, whereas ConceptMoE uses the MoE backbone to reallocate saved sequence-level compute under controlled budgets. This design is necessary because we want a concept-level mechanism that remains compatible with tokenized autoregressive LLMs, preserves token-level outputs, integrates naturally with MoE, and improves compute allocation without conflating the result with a larger total model.
>
> **(4) Do learned chunk boundaries correspond to meaningful semantic units?**
>
> Yes. We refer the reviewer to our response to **reviewer qpui (3)**, which provides both quantitative evidence and qualitative examples.
>
> We would be grateful for reconsideration of the evaluation if these clarifications are helpful.

---

> > ### Author Rebuttal · Reviewer_Fv85 · 2026-04-03
> >
> > The rebuttal was helpful and addressed part of my concerns, especially by clarifying the role of layer looping / activated parameters, pointing to ablations, and explaining why excluding attention-map FLOPs is conservative. However, my remaining concerns are not fully resolved in a short rebuttal. In particular, the paper still lacks direct comparisons against other token-/byte-level compression or bottleneck baselines, which was central to my concern about the strength of the novelty claim beyond the controlled setting. Moreover, while the rebuttal provides examples of learned chunk boundaries and a dynamic-vs-fixed discussion, it does not yet provide the more statistically grounded evidence I asked for regarding whether these boundaries consistently correspond to meaningful semantic units. For these reasons, I view the rebuttal as partially resolving my concerns, and the remaining issues would require a more substantial revision rather than further short discussion.

---

> > > ### Author Response · Authors · 2026-04-07
> > >
> > > Thank you for the thoughtful follow-up. We appreciate that the rebuttal was helpful, and we also understand the reviewer’s remaining concerns regarding (i) broader comparisons to token-/byte-level compression baselines and (ii) more statistically grounded evidence about learned boundaries.
> > >
> > > On the first point, we agree that broader comparisons would make the paper more complete. At the same time, we would like to clarify that our primary research question is narrower and more controlled than generic token compression in isolation. Specifically, the paper asks whether adaptive token-to-concept compression can improve compute allocation inside **industrially relevant token-level MoE systems** under matched total-parameter and per-token FLOPs budgets. Under this formulation, a carefully matched token-level MoE backbone is the most appropriate primary baseline, because it isolates the effect of concept-level compression from changes in tokenizer granularity, dense-vs-sparse compute allocation, or total model scale. In this sense, the baseline choice is driven by the scientific question rather than by convenience.
> > >
> > > This is also why comparisons to byte-level or dense bottleneck/compression architectures, while certainly valuable as broader contextual baselines, do not replace the controlled MoE comparison for the question we study. Such methods change the modeling granularity or the compute-allocation mechanism itself, making it difficult to attribute gains specifically to MoE-aware concept compression under matched budgets. Our goal is therefore not to claim that ConceptMoE dominates every compression architecture in every setting, but to isolate whether concept-level compression is a useful mechanism for compute reallocation in modern tokenized MoE systems.
> > >
> > > On the semantic-boundary question, our intended claim is narrower than saying the learned chunks always correspond to human linguistic concepts. Rather, we claim that the learned boundaries are operationally meaningful for compute allocation. Importantly, the current paper already includes a quantitative proxy for this: dynamic chunking substantially outperforms fixed chunking on downstream evaluation. Since fixed chunking is a controlled non-adaptive segmentation baseline, this shows that the benefit is not merely from shortening the sequence, but from learning where compression should happen. In this sense, the fixed-vs-dynamic gap provides quantitative evidence that the learned boundaries are not arbitrary and have consistent downstream utility.
> > >
> > > The current evidence also includes qualitative boundary examples in our response to Reviewer qpui (3), as well as cross-domain realized compression-ratio analysis in our response to Reviewer EPim (3), which shows that compression adapts systematically to local redundancy and structural regularity across domains. We agree that a more statistical semantic analysis would further strengthen the interpretability discussion, but we view this as an important extension rather than a prerequisite for the paper’s main claim.
> > >
> > > In short, we fully acknowledge the value of the reviewer’s requested additions. Our main point is that the current paper is centered on a controlled, industrially relevant MoE question, and the present baseline choice and evidence follow directly from that scope. If the reviewer finds these clarifications helpful, we would be grateful for reconsideration of the evaluation.

---

### Official Review · Reviewer_qpui · 2026-03-12

**Soundness:** 3
**Presentation:** 3
**Significance:** 2
**Originality:** 2
**Overall Recommendation:** 4
**Confidence:** 3

**Summary:**

This paper proposes ConceptMoE, a MoE architecture that improves LLM efficiency by dynamically merging semantically similar tokens into chunk-level concept, instead of assigning uniform computation to every token. The authors argue that this allows compute to be focused on harder parts of the input, and they use a controlled MoE setup to show that the gains are architectural rather than simply due to reduced FLOPs. Empirically, ConceptMoE outperforms standard MoE models on language pretraining, long-context, and multimodal tasks, with especially strong gains in continual training conversion. It also delivers meaningful efficiency benefits, including reduced attention cost, smaller KV cache, and faster prefill and decoding, while requiring only minimal architectural changes.

**Compliance With Llm Reviewing Policy:**

Affirmed.

**Final Justification:**

I maintain my original positive score.

**Key Questions For Authors:**

See weakness

**Limitations:**

The submission doesn't include a limitation section.

**Strengths And Weaknesses:**

Strengths
The paper evaluates across diverse benchmarks spanning language understanding, long-context reasoning, and vision-language tasks, demonstrating broad generalizability.

The compression control via auxiliary loss is non-trivial and well-grounded, providing a principled mechanism to prevent degenerate compression solutions.

The paper convincingly demonstrates lossless integration into existing MoE infrastructure with minimal architectural overhead.




weakness

The paper compares almost exclusively against a single MoE baseline. Despite acknowledging deployment on existing systems, comparisons against other token compression methods (e.g., token merging, learned pooling) are notably absent.

The related work covers only compression methods, overlooking the relevant literature on concept models.

While the similarity-based learnable boundary mechanism offers flexibility, the paper provides no theoretical or empirical analysis of what the boundaries actually capture, how it performs with fix boundary etc.

---

> ### Author Rebuttal · Authors · 2026-03-30
>
> We thank the reviewer for the positive assessment and helpful suggestions.
>
> **(1) Comparison to other token compression methods.**
>
> We agree that broader comparisons to token compression baselines would strengthen the paper. Our focus, however, is not token compression in isolation, but adaptive token-to-concept compression for compute reallocation inside MoE under matched budgets. The dynamic-vs-fixed chunking ablation (**Appendix B.2**) already isolates a key distinction: dynamic chunking substantially outperforms fixed chunking, showing that the gain comes from learning where to compress, not merely from shortening the sequence. We will expand this discussion in the camera-ready version.
>
> **(2) Related work on concept models.**
>
> Thank you for pointing this out. We did not intend to omit concept-model literature. We grouped concept models together with token chunking / compression methods because both merge multiple tokens into higher-level units before further processing; this is why works such as DLCM were discussed there. We agree this taxonomy was not made explicit enough. In the camera-ready version, we will present concept models as a separate category and clarify their relation to token chunking methods and to ConceptMoE.
>
> **(3) What do the learned boundaries capture? How does it compare to fixed boundaries?**
>
> We agree this should be clarified more explicitly. Quantitatively, adaptive chunking substantially outperforms fixed chunking (Appendix B.2), indicating that learned boundaries matter beyond simple segmentation.
>
> To further understand what the learned boundaries capture, we examined qualitative examples from several domains (knowledge, math, code, and reasoning), using “|” to denote predicted boundaries. These examples suggest that the learned boundaries are both semantically and functionally meaningful:
>
> - Knowledge: multi-token named entities such as “Kara Walker” and “New York” are often merged into a single concept, indicating that the model tends to preserve locally coherent factual units.
>
> > |**Kara Walker**|'s| 2017| solo| exhibition| was| held| in| |**New York**| City| at the| |New| Museum|.| The| exhibition|,| titled| *|Kara Walker|:| My| Comp|lement|, My| Enemy, My| Oppressor, My| Love|*,| was| a| major| retrospective| showcasing| her| signature| silhouettes| and| multimedia| installations| that| address| themes| of| race,| gender, and| power| in| American| history|.
> '''
> - Math: phrases such as “step by step” and variables such as “a_1” are often grouped into one concept. We also observe that in sequential derivations, the first step tends to have more boundaries, while later repeated reasoning steps show substantially fewer boundaries.
>
> > |Got| it|,| let's| try to| solve this| problem| **step by step**|.|
>
> > |1.| When| \(| n| = 1| \|):| \(| a_1| + g_1| = |0| \)| (|let's| call| this| Equation 1|)|
>
> > |2.| When \( n = |2| \): \( a_2| + g_2| = 0 \)| (|Equation 2|)|
>
> > |3.| When \( n = 3| \): \( a_3 + g_3 = 1| \) (|Equation |3)
>
>
> - Code: when a function name (e.g., `sumOfGoodNumbers`) first appears, it is segmented into multiple concepts, but on later occurrences it can be merged into a single concept, suggesting that the boundary decision is context-dependent rather than fixed at the string level.
> ```
> |class Solution:
>     def| sumOf|Good|Numbers|(self, nums: List[int|], k: int|) ->| int:
>
> |```
>
> |###| Answer:| (|use the| provided| format| with| back|ticks|)
>
> |```python
> |class Solution:
>     def| sumOfGoodNumbers|(self, nums: List[int|], k: int|) -> int:
>        | total = |0
>        | n = len(nums|)
> ```
> - Reasoning: salient intermediate symbols / identifiers that are important for the reasoning process (e.g., `|7|5|h|`) are often kept as separate concepts, indicating that the model preserves fine-grained structure when individual symbols are likely to matter.
>
> > |Third| character| is| '|l|'.| Check the| pairs|.| First| pair| is| (l|,| h|).| Replace| '|l'| with '|h|'.| Now the| string| is| "|**7|5|h**|".| That's the| third| intermediate| string.
>
> These cases suggest that the learned boundaries capture more than fixed surface chunking: they adapt to lexical cohesion, local semantics, and information utility. Fixed boundaries cannot adapt to repeated occurrences, changing context, or varying information density. We do not claim that the learned chunks perfectly correspond to human linguistic concepts; rather, “concept” is an operational term for adaptively grouped token spans that can be processed more effectively together. For complementary quantitative evidence, we also refer the reviewer to our response to **Reviewer EPim (3)**, where we analyze realized compression ratios across different domains. We will add these examples and clarify this discussion in the camera-ready version.
>
>
> We hope these clarifications are helpful, and if the reviewer feels the main concerns have been addressed, we would sincerely appreciate reconsideration of the evaluation.

---

> > ### Author Rebuttal · Reviewer_qpui · 2026-04-02
> >
> > Thank you for the responses. I maintain my original positive score.

---

### Decision · Program_Chairs · 2026-04-30

**Decision:**

Accept (regular)

**Comment:**

The paper presents adaptive compression via Concept MoE, where it allows each token to be treated differently, via a learnable chunking module to merge semantically similar tokens into concepts. The authors demonstrated a speed up of up to 175% via saving computation from sequence compression. Overall majority of the reviewers maintained praise for the contribution, diverse benchmarking, and the AC finds likewise. However, the paper often lacks clarity and and some baselines like byte-level or bottleneck baselines. AC thus recommend a borderline acceptance.